# RSM- and ANN-Based Multifrequency Ultrasonic Extraction of Polyphenol-Rich *Sargassum horneri* Extracts Exerting Antioxidative Activity via the Regulation of MAPK/Nrf2/HO-1 Machinery

**DOI:** 10.3390/antiox13060690

**Published:** 2024-06-04

**Authors:** Ahsan Javed, Md Badrul Alam, Marufa Naznin, Raees Ahmad, Chang Hyung Lee, Sunghwan Kim, Sang-Han Lee

**Affiliations:** 1Department of Food Science and Biotechnology, Graduate School, Kyungpook National University, Daegu 41566, Republic of Korea; ahsanjaved@knu.ac.kr (A.J.); mbalam@knu.ac.kr (M.B.A.); 2Food and Bio-Industry Research Institute, Inner Beauty/Antiaging Center, Kyungpook National University, Daegu 41566, Republic of Korea; 3Department of Chemistry, Kyungpook National University, Daegu 41566, Republic of Korea; naznin@knu.ac.kr (M.N.); ahmed@knu.ac.kr (R.A.); 4Mass Spectroscopy Converging Research Center, Green Nano Materials Research Center, Kyungpook National University, Daegu 41566, Republic of Korea; 5Bio-MAX Institute, Seoul National University, Seoul 08826, Republic of Korea; changhyung@snu.ac.kr

**Keywords:** *Sargassum horneri*, multifrequency ultrasonic-assisted extraction, secondary metabolites, antioxidantive activity

## Abstract

*Sargassum horneri* (SH) is widely consumed as a healthy seaweed food in the Asia–Pacific region. However, the bioactive components contributing to its biological activity remain unknown. Herein, we optimized multifrequency ultrasonic-assisted extraction conditions to achieve higher antioxidant activity using a response surface methodology and an artificial neural network. High-resolution mass spectrometry (HRMS; negative mode) was used to tentatively identify the secondary metabolites in the optimized SH extract, which were further tested against oxidative stress in RAW264.7 cells. Additionally, the identified compounds were analyzed in silico to determine their binding energies with the Keap1 protein (4L7B). We identified 89 compounds using HRMS, among which 19 metabolites (8 polyphenolics, 2 flavonoids, 2 lignans, 2 terpenes, 2 tannins, 2 sulfolipids, and 1 phospholipid) were putatively reported for the first time in SH. The in vitro results revealed that optimized SH extract inhibited oxidative stress via the Nrf2/MAPKs/HO-1 pathway in a dose-dependent manner. This result was validated by performing in silico simulation, indicating that sargaquinoic acid and glycitein-7-*O*-glucuronide had the highest binding energies (−9.20 and −9.52 Kcal/mol, respectively) toward Keap1 (4L7B). This study offers a unique approach for the scientific community to identify potential bioactive compounds by optimizing the multivariant extraction processing conditions, which could be used to develop functional and nutraceutical foods.

## 1. Introduction

Reactive oxygen and nitrogen species (ROS/RNS) play a significant role in maintaining cellular homeostasis, and an imbalanced redox status or uncontrolled ROS/RNS levels induce the production of numerous oxidative stress-linked ailments, including cancer, inflammation, and cardiovascular diseases [1]. Hydrogen peroxide (H_2_O_2_) is a widely used inducer that triggers oxidative damage and stress as it undergoes Fenton’s reaction with Fe^2+^ ions, leading to the formation of a highly reactive -OH radical. Although macrophages defend cell homeostasis from numerous infectious agents, they produce ROS and RNS upon stimulation, causing epigenetic alterations that contribute to the development of chronic illness [2]. Therefore, activated macrophage models are widely used to propose functional diets through a multi-targeted strategy. Phytochemicals with intrinsic antioxidant activity can directly or indirectly activate a wide range of protective signaling cascades and may be employed to treat oxidative stress-related diseases [3]. As a result, understanding and confirming natural chemical actions, as well as identifying the underlying molecular pathways, are critical for determining their potential medicinal relevance.

Many antioxidant enzymes and detoxifying agents, especially heme oxygenase-1 (HO-1) are relied upon in the induction of nuclear factor erythroid 2-related factor 2 (Nrf2) [4]. Under normal conditions, cytosolic Kelch-like ECH-associated protein 1 (Keap1) causes the degradation of Nrf2 through the ubiquitin–proteasome system. Oxidative stress or xenobiotic challenge can prevent Nrf2 degradation by modifying the reactive cysteine residue of Keap1, leading to the translocation of Nrf2 to the nucleus and its binding to antioxidant-related elements (AREs) in the promoter regions of antioxidant and cytoprotective genes [5]. Furthermore, Nrf2 nuclear translocation is dependent on the activation of mitogen-activated protein kinases (MAPKs), phosphatidylinositol 3-kinase/Akt (PI3K/AKT), and protein kinase C (PKC) [6]. *Sargassum horneri* (SH) is a widely used foodstuff in the Asia and Pacific region due to its well-known attributes. It possesses numerous bioactive ingredients including polyphenols, polysaccharides, lignans, and terpenes, reflecting its antimicrobial, antioxidative, antidiabetic, and anticancer characteristics as a food resource. So far, various marine bioactive ingredients have been isolated from SH and studied against different diseases such as diabetes, cancer, and neurological disorders [7]. Additionally, Jayawardena et al. [8] demonstrated that ethanolic SH extracts helped attenuate the fine dust-caused inflammation. Likewise, Han et al. [9], isolated the (−)-lolidolide and determined its anti-inflammatory activity, which involves activating the Nrf2/HO-1 signaling in α-MSH-stimulated HaCaT cells. 

In any experiment, the first and most crucial phase is the recovery and purification of bioactive compounds. Artificial neural networks (ANNs) and response surface methodology (RSM) are widely used statistical tools for the optimization of the multivariant parameter to replace non-conventional methods; they also provide the predicted results, thus helping the optimization [10,11,12]. Likewise, in silico approaches, especially molecular docking simulation, are gaining the attention of the scientific community to screen out receptor–ligand interactions in the suppression or activation of specific proteins [13]. To the best of our knowledge, the multifrequency ultrasonic extraction optimization of antioxidant activity using two sophisticated statistical tools (ANNs and RSM), the profiling of secondary metabolites using high-resolution mass spectrometry (HRMS), and the quantification of the binding energies of identified compounds using molecular docking simulation have never been performed. Our study focused on optimizing the extraction conditions to achieve maximum antioxidant activity and profiling the bioactive compounds responsible for those biological activities using HRMS. Additionally, in vitro and in silico processes were conducted to attenuate the oxidative stress via Nrf2/MAPKs/HO-1 cascade using the RAW264.7 cells and determine the binding energies of the potentially identified compounds using molecular docking.

## 2. Materials and Methods

### 2.1. Sample Collection and Preparation

In the middle of April 2021, *Sargassum horneri* (SH) samples were collected from the coastal areas of the Republic of Korea and verified by the scientific officer at the Department of Oceanography, Kyungpook National University, Republic of Korea. Samples were washed and dried at 37 °C, and multifrequency ultrasonic-assisted extraction (MUAE) was performed as briefed by Vázquez-Rodríguez et al. [13] with slight modifications. Briefly, powdered SH (10 g) was soaked for 1 h before extraction with 150 mL of solvent according to the RSM experimental design Table 1. The sample-to-solvent ratio was 1:15 for MUAE. Based on the RSM–Box–Behnken design (BBD) model, MUAE was performed using an ultrasonic bath (Elma Schmidbauer GmbH, Singen, Germany) at 26, 35, and 40 kHz [14]. The MUAE samples were kept at 25 °C for 30 min to increase the extractability of bioactive components, filtered (Whatman No. 1; Schleicher & Schuell, Keene, NH, USA), and lyophilized using a freeze dryer (Il-Shin Biobase, Goyang, Republic of Korea). Samples were then stored at −20 °C in the polyethylene bags for further analysis.

### 2.2. Experimental Designs Using Statistical Models

The RSM–BBD model followed the methodology of Javed et al. [15]. In short, four independent variables—time (X_1_), temperature (X_2_), ethanol concentration (X_3_), and ultrasonic frequency (X_4_)—were checked against four experimental responses—total phenolic content (TPC, Y1), total flavonoid content (TFC, Y_2_), 1-diphenyl-2-picrylhydrazyl (DPPH, Y_3_), and 1-diphenyl-2-picrylhydrazyl (ABTS, Y_4_). ANOVA and quadratic polynomial equations were used to calculate the significance level of the fitted model. After developing the RSM statistical model, the ANN model was constructed using experimental values to check the accuracy of the two models. The ANN model conditions were developed by following the methodology illustrated by Javed et al. [16]. In short, ANNs comprised hidden layers (tan and purplin) with cascade-forward and feed-forward networks. Two different functions, that is, Levenberg–Marquardt back-propagation (trainLM) and Broyden–Fletcher–Goldfarb–Shanno (BFGS) were applied to construct the model using deep learning toolbox in MATLAB R2020a (MathWorks, Natick, MA, USA). For accurate model construction, 65% of experimental data was used for input, 20% of experimental data was used for testing, and 15% of experimental data was used for validation. The hit and trial method was applied to minimize the mean square error, MSE. After constructing the RSM and ANN models, both models were compared using 4 different parameters elaborated by Javed et al. [15], that is, the absolute average deviation (AAD), the standard error of prediction (SEP), the coefficient of determination (R^2^), and the root mean square error (RMSE). To calculate the optimized condition for the ANN model, a further genetic algorithm (GA) was applied. In the GA model, ANN data were used as a fitness function, and the highest results for the target variables were determined. The mutation function, initial population size, crossover percentage, and evolutionary algebra were selected based on the current situation while the other parameters of GA were kept at their default settings (Appendix A). 

### 2.3. Antioxidant Activities

SH extracts were analyzed thrice for their TPC and TFC using the Folin–Ciocalteu test and the aluminum chloride colorimetric method, respectively, by following the method described by Javed et al. [15]. Regression equations (TPC: y = 0.0526x + 0.0020; R^2^ = 0.9856, TFC: y = 0.0 336x + 0.0021; R^2^ = 0.9894) were derived from calibration curves. TPC and TFC were expressed in gallic acid and catechin equivalents ((mg)/dry weight sample (g)), respectively. ABTS and DPPH radical scavenging assays were analyzed using ascorbic acid as a standard as per Alam et al. [3]. The calibration curves were used to calculate the regression equations for DPPH (y = 0.0555x + 0.0123; R^2^ = 0.9879) and ABTS (y = 0.0369 + 0.0133; R^2^ = 0.9823). 

**Table 1 antioxidants-13-00690-t001:** Comparison of Box–Behnken design (BBD) for independent variables against corresponding target responses (experimental), RSM and ANN predicted values.

Run No	Parameter				TPC (mg GAE/g)			TFC (mg CAE/g)			DPPH (% Inhibition)			ABTS (% Inhibition)		
	X_1_ (%)	X_2_ (min)	X_3_ (°C)	X_4_ kHz	Experimental Value	RSM Predicted	ANN Predicted	Experimental Value	RSM Predicted	ANN Predicted	Experimental Value	RSM Predicted	ANN Predicted	Experimental Value	RSM Predicted	ANN Predicted
**1**	50	50	20	35	47.78 ± 3.25	49.27	49.55	35.77 ± 1.52	38.84	35.39	19.24 ± 1.62	20.10	19.26	33.07 ± 1.19	32.65	33.09
**2**	50	35	60	26	55.85 ± 3.54	54.93	58.95	20.08 ± 3.14	24.99	20.06	23.58 ± 1.78	22.85	22.63	35.01 ± 1.74	30.84	35.54
**3**	50	20	40	40	57.62 ± 2.87	60.47	60.48	23.49 ± 2.87	24.88	23.61	26.59 ± 2.17	25.46	26.74	32.11 ± 1.76	34.76	32.92
**4**	50	50	60	35	71.80 ± 1.98	71.23	71.96	36.54 ± 2.36	34.91	38.89	30.36 ± 1.15	29.09	29.69	32.01 ± 1.66	34.34	32.16
**5**	30	50	40	35	65.33 ± 4.25	67.46	67.04	22.04 ± 3.14	23.66	27.68	25.65 ± 2.17	25.54	25.69	49.29 ± 1.39	45.89	49.31
**6**	50	50	40	40	66.40 ± 2.64	61.25	64.13	32.71 ± 2.77	33.02	32.96	26.04 ± 1.94	23.66	24.69	32.16 ± 1.58	33.16	32.43
**7**	50	50	40	26	65.33 ± 3.14	64.58	64.30	28.57 ± 3.35	23.77	28.84	25.65 ± 1.81	26.75	25.69	38.04 ± 1.95	38.24	37.65
**8**	50	35	60	40	65.95 ± 1.64	66.07	64.15	35.81 ± 2.31	33.71	35.46	26.33 ± 0.37	26.31	25.37	33.79 ± 2.75	31.37	33.11
**9**	70	50	40	35	53.03 ± 2.88	55.90	55.18	57.45 ± 2.87	58.90	59.83	17.46 ± 1.75	19.28	15.56	20.03 ± 3.88	20.35	18.57
**10**	30	35	40	40	63.56 ± 4.35	63.72	63.08	22.04 ± 2.70	19.80	24.23	23.58 ± 1.44	24.24	23.82	43.83 ± 3.65	42.28	42.29
**11**	50	35	40	35	66.25 ± 1.61	67.37	67.87	25.55 ± 1.93	26.78	30.05	28.19 ± 1.52	29.34	28.38	44.22 ± 2.44	44.87	43.17
**12**	30	20	40	35	67.65 ± 3.47	65.86	65.63	22.67 ± 3.40	23.95	22.34	26.01 ± 1.85	25.18	25.24	38.21 ± 3.36	35.48	39.26
**13**	30	35	60	35	64.40 ± 2.95	66.65	65.95	24.76 ± 1.88	27.94	24.88	21.68 ± 1.17	24.44	21.63	34.40 ± 2.38	37.80	34.26
**14**	30	35	40	26	69.76 ± 1.98	68.74	70.00	22.87 ± 2.67	20.52	25.6	31.06 ± 1.99	29.73	31.05	32.27 ± 2.35	34.60	31.39
**15**	50	20	60	35	58.12 ± 3.16	55.24	58.67	35.92 ± 3.79	33.78	34.02	20.91 ± 2.33	19.38	20.16	29.95 ± 2.01	30.32	29.34
**16**	50	20	40	26	50.57 ± 2.75	54.26	52.15	28.22 ± 3.02	24.01	28.7	15.95 ± 1.34	17.74	15.03	27.22 ± 1.58	28.33	25.49
**17**	70	20	40	35	50.17 ± 2.97	49.13	49.92	47.20 ± 2.75	48.33	48.85	14.40 ± 2.20	15.52	14.6	24.73 ± 2.98	25.73	25.71
**18**	70	35	40	26	47.69 ± 3.11	46.29	47.69	41.30 ± 1.79	42.90	41.44	13.06 ± 1.12	11.74	12.79	24.24 ± 3.45	25.97	22.54
**19**	30	35	20	35	61.70 ± 3.77	60.00	62.25	27.95 ± 2.78	26.50	26.29	28.75 ± 2.30	27.62	28.86	36.34 ± 2.15	38.31	35.5
**20**	50	35	40	35	68.01 ± 2.58	67.37	67.87	26.78 ± 2.35	26.78	30.05	31.41 ± 1.60	29.34	28.38	45.85 ± 2.64	44.87	43.17
**21**	50	20	20	35	57.71 ± 3.79	56.89	58.75	27.12 ± 1.97	29.68	27.4	25.08 ± 1.89	25.69	23.98	34.02 ± 2.78	31.64	31.77
**22**	50	35	20	26	56.82 ± 4.20	57.25	57.09	24.74 ± 2.78	29.61	26.32	22.46 ± 2.64	22.99	23.22	31.65 ± 2.25	30.47	33.00
**23**	70	35	40	40	54.69 ± 3.13	54.19	53.26	48.86 ± 1.95	53.73	48.9	21.18 ± 1.52	21.85	20.19	22.24 ± 2.46	19.62	22.75
**24**	70	35	60	35	53.98 ± 2.65	56.01	54.85	58.59 ± 2.60	56.39	55.7	20.18 ± 1.39	21.00	20.03	20.33 ± 2.62	20.85	18.74
**25**	70	35	20	35	44.28 ± 2.95	42.35	46.88	64.49 ± 3.10	57.66	62.51	18.22 ± 2.29	15.14	17.55	20.88 ± 2.35	19.97	19.10
**26**	50	35	20	40	46.44 ± 3.57	48.98	48.97	33.21 ± 3.16	31.01	30.82	21.93 ± 1.67	24.15	21.26	28.34 ± 3.69	31.29	31.17
**27**	50	35	40	35	67.85 ± 4.10	67.37	67.87	28.02 ± 1.30	26.78	30.05	28.41 ± 1.71	29.34	28.38	44.55 ± 2.45	44.87	43.17

### 2.4. High-Resolution Mass Spectroscopy (HRMS)

Electrospray ionization tandem mass spectrometry (ESI-MS/MS) analysis was conducted as per the procedures of Choi et al. [17]. In brief, optimized SH extracts were injected using (15 μL/min) into the ESI of the Q-Exactive Orbitrap mass spectrometer (Thermo Fisher Scientific Inc., San Jose, CA, USA) using a 500 μL graduated syringe. The setting of the ESI-MS was as follows: the mode was selected as negative; full width at half maximum was kept at a mass resolution of 140,000; the flow rates of sweep gas and sheath gas were kept at 0 and 5, respectively, whereas the temperature was kept at 320 °C and the spray voltage was set at 4.20 kV. For a parent peak, fragmentations were conducted at three different collision energies (CEs = 10, 30, and 40). The other condition of the MS/MS experiment was kept the same except for the auxiliary and sheath gas rates, which were 10, respectively, and an S-lens Rf level of 50. Xcalibur 3.1 and Foundation 3.1 (Thermo Fisher Scientific Inc., Rockford, IL, USA) were used to analyze and process peak identification and spectral data. Fragments obtained from the negative mode were verified by comparing with the online databases, i.e., METLIN, HMDB, and FoodB.

### 2.5. Cell Viability Assay

Cell viability was assessed via MTT assay, following Alam et al. [5]. Briefly, H_2_O_2_ pretreated Raw264.7 cells (American Type Culture Collection, Manassas, VA, USA) were treated with various concentrations of optimized SH extracts (5–20 mg/mL) and gallic acid (50 µg/mL). Cells were incubated for 18 h at 37 °C and media was removed using suction. Cells were incubated again for 30 min after treatment with 100 µL of MTT (10%) solution. After the suction of the MTT solution, 100% dimethyl sulfoxide (DMSO) was added, and the optical density was read at 590 nm using a microplate reader (Victor3, PerkinElmer, Waltham, MA, USA).

### 2.6. Reactive Oxygen Species (ROS) Analysis

ROS produced as an outcome of oxidative damage were analyzed using the methodology of Alam et al. [5], using the DCFH-DA procedure. In short, seeded RAW 264.7 cells were pretreated with H_2_O_2_, incubated for 30 min, and treated with optimized SH extracts (5–10 mg/mL) and GA (50 µg/mL). Raw264.7 cells were washed with PBS, treated with DCFH-DA (25 µM), and incubated for 30 min at 37 °C. The fluorescence intensity was analyzed at 485 nm (excitation) and 528 nm (emission) using a Victor fluorescence microplate reader (PerkinElmer) to calculate the ROS generation rate.

### 2.7. Western Blotting and Cell Lysate Preparation

After treatment, cells were disintegrated with RIPA buffer to collect the proteins. The collected proteins were then treated with 5X SDS-PAGE (3M Science, Seoul, Republic of Korea) sample buffer and denatured at 95 °C for 12 min. Proteins were analyzed by 10% SDS gel electrophoresis for 95 min, and protein bands were transferred onto nitrocellulose membranes for 2 h. Skim milk (5%) and bovine serum albumin (5%) were added to the membrane, which was then incubated overnight at 4 °C with a first antibody. After incubation, membranes were washed and treated with the secondary antibody (anti-goat and anti-rabbit IgG). Bands were detected using a chemiluminescence system (PerkinElmer) and identified based on their molecular weights [5].

### 2.8. Ligand Retrieving and Protein Preparation

The structures of putatively identified compounds were retrieved online using the PubChem database. The energy minimization of all the structures and Keap1 (4L7B) protein was performed with the Chimera software (version 1.5), using the procedure of Diniyah et al. [18].

### 2.9. Molecular Docking Study

Among the 89 identified compounds, 55 compounds (phenolic, flavonoids, terpenes, lignans, and tannins) were used to analyze their binding affinity with Keap1 protein (4L7B) using Auto-Dock Vina 4.2.621 through DockingApp’s interface. All the water molecules and ligands were removed from the initial structure and polar hydrogen atoms were added before the docking. After carefully analyzing the literature, we made the size of the grid box 25 Å × 25 Å × 25 Å with its center at position x = 2.4, y = 2.8, and z = −29.21. ΔG values were calculated to measure each compound’s root-mean-square deviation (RMSD) [18]. Compounds with an RMSD of ≤9.0 were further compared with the MolDock scores using MolDock software (version 7.0.0). The same grid box information was used with the MolDock optimizer algorithm to obtain the MolDock score. Pymol and Discovery Studio were used to visualize the 2D and 3D interaction between the protein amino acid and ligands.

### 2.10. Statistical Analysis

The experimental results were statistically analyzed using Design Expert 11 (version 8.0.6, STAT-EASE, Inc., Minneapolis, MN, USA), MATLAB R2020a software (MathWorks), and OriginPro software (OriginPro^®^ 2021b SR1 v9.8.5.204). All results are reported as the mean standard deviation of three independent experiments (*n* = 3), with at least three replicates for each sample in each experiment. The value of *p* < 0.05 was considered statistically significant, and different letters represent statistically different means.

## 3. Results

### 3.1. Design Matrix and Fitting RSM Modeling

Optimizing the UAE technique using different frequencies in this experiment aimed to increase the extractability of bioactive ingredients with minimum resource utilization. The experimental values for the 27 runs are listed in Table 1. The target responses of experimental UAE exhibited various values for TPC (44.28–71.80 mg GAE/g), TFC (20.08–64.49 mg CAE/g), DPPH activity (13.063–31.41% inhibition), and ABTS activity (20.03–49.29% inhibition) against independent variables (X_1_, X_2_, X_3_, and X_4_). The model fitness can be determined using the model significance and the lack of fitness, which should be non-significant, and, in both cases, our model fulfills the requirement of the well-fitted model as shown in Appendix A. The RSM- and ANN-predicted models were significant to the target response model in Table 1; although, when compared to each run, various targeted responses are non-significant to the predicted RSM and ANN model run numbers (Table 1). Similarly, in the RSM model, for TPC, the concentration, time, and temperature showed significant effects, whereas, in the case of TFC and ABTS, only the concentration showed a significant effect. For DPPH, concentration and time exhibited a significant trend (Appendix A). Likewise, for TPC, the interactions of time × temperature and temperature × frequency and the squares of temperature, time, concentration, and frequency also exhibited a significant effect. In the case of TFC, the squares of concentration and temperature showed a significant effect in the RSM model. For DPPH, the interactions of concentration × frequency and time × temperature and the squares of concentration and time exhibited a significant effect. In ABTS, a significant effect was observed in the squares of concentration, time, temperature, and frequency. 

In this study, the R^2^ values of each MUAE value of all target responses (R^2^ = 0.91–0.93) were higher than the acceptable range (R^2^ ≥ 0.80), indicating that the model is well fitted as observed in Appendix A. Likewise, adequate precision represents a signal-to-noise ratio, and higher than four is desirable for a well-fitted model. In the current study, the ratio was between 11.04 and 13.04, suggesting that the model was well fitted and had appropriate signals to traverse the design space. 

RSM contour plots were created to better understand the relationship between the independent and target response variables (Figure 1A–D). The highest values for TPC (71.80 mg GAE/g) and TFC (64.49 mg CAT/g) were observed in run no. 4 (X_1_ = 50%, X_2_ = 50 min, X_3_ = 60 °C, and X_4_ = 35 kHz) and run no. 25 (X_1_ = 70%, X_2_ = 35 min, X_3_ = 20 °C, and X_4_ = 40 kHz), respectively. A DPPH value was achieved in run no. 20 (X_1_ = 50%, X_2_ = 35 min, X_3_ = 40 °C, and X_4_ = 35 kHz), and the maximum ABTS value was achieved in run no. 5 (X_1_ = 30%, X_2_ = 50 min, X_3_ = 40 °C, and X_4_ = 35 kHz), respectively.

Figure 2 illustrates the optimal ANN architecture topology for UAE conditions. The hit and trial method was adopted to minimize the mean error, and an ANN topology of 4–10–4 was sufficient to optimize the ANN model as the neuron optimization was conducted on up to 20 neurons to find the best fitness values (lower MSE). The results indicated that, for all the dependent variables, the lowest MSE value was calculated for ten neurons, which was then further checked via the validation performance and histogram error as depicted in Figure 2. 

Appendix A shows the comparison in terms of the prediction abilities and the ANN model reflected better predicting ability than RSM. Appendix A exhibits the optimized model conditions for both the RSM and ANN models. After optimizing the condition, we repeated the experiment with slight parameter changes (X_1_: 50%, X_2_: 30 min, X_3_: 35 °C, and X_4_: 35 kHz) to validate the optimized results. The optimized SH extract exhibited TPC (66.34 mg GAE/g), TFC (43.63 mg CAT/g), DPPH (29.11% of inhibition), and ABTS (46.77% of inhibition), respectively. The validation condition also showed that the ANN-optimized model conditions are better in terms of prediction.

### 3.2. SH Extract Metabolite Profiling Using High-Resolution LC-MS/MS

Secondary metabolites in the UAE-optimized SH extracts were further tentatively identified using the negative mode of ESI-MS/MS equipment. Table 2 indicates the presence of 89 compounds, including 19 metabolites (8 phenolics, 2 flavonoids, 2 lignans, 2 terpenes, 2 tannins, 2 sulfolipids, and 1 phospholipid) that were tentatively reported for the first time in SH-negative mode using MS data from the parent ion mass, recognized fragmentation patterns for the given classes of compounds, and neutral mass loss, paired with comparisons of the existing literature and searches in online databases.

#### 3.2.1. Phenolic Acids and Flavonoids 

The literature showed that phenolic acids lose certain fragments such as methyl (15 Da), hydroxyl (18 Da), and carboxyl (44 Da) during the collision, which helped in determining the compounds [3]. Compounds **1**–**14** have been tentatively identified in *Sargassum fusiforme* based on previously reported fragmentation behavior data (Table 1) [15,16,17,18,19]. Interestingly, 5-(3′,5′-Dihydroxyphenyl)-γ-valerolactone 3-*O*-glucuronide and coumaroylquinic acid were tentatively acknowledged for the first time in optimized MUAE SH extract. Compounds **15**–**17** were previously reported in *Centroceras* sp., *Ecklonia* sp., and *U. fasciata* [20]. These compounds were tentatively identified for the first time in the SH. Compounds **18**–**22** were previously reported in *Dasya* sp., *Centroceras* sp., and *Sargassum* sp., respectively [21,22,23]. Compounds **21** and **22** putatively reported for the first time in the SH.

Each subgroup of polyphenols has a distinct fragmentation behavior in MS/MS, including flavonoids. Typically, C-ring bonds cleave (RDA process), resulting in A or B rings, and sometimes portions of the C ring [5]. Similarly, a few small neutral ions, such as CO (−28 Da), C_2_H_2_O (−42 Da), CO_2_ (−44 Da), and 2CO (−56 Da), might also be yielded during the disintegration. In the previous literature, depending on the fragmentation behavior, compounds **23**–**28** were identified as flavonoid glycosides epigallocatechin, epicatechin, dalbergin, catechin, dihydrobiochanin A, and glycitein 7-*O*-glucuronide. Dihydrobiochanin A (**27**) was previously discovered in *Grateloupia* sp., *Ascophyllum nodosum*, green seaweeds, and *Ecklonia* sp., respectively. Likewise, glycitein 7-*O*-glucuronide (**28**) was reported in *Centroceras* sp., but both compounds were putatively identified in SH for the first time in this study (Table 2).

#### 3.2.2. Lignans and Terpenes

Few main fragmented ions of [M–H-15], [M–H-18], [M–H-44], and [M–H-46] were produced in dibenzylbutyrolactones as a result of a loss of CH_3_, H_2_O, or CO_2_, and also in the combination of CO and H_2_O. However, butanediol lignans generate [M–H-48] ions by breaking at the β-position as a combined loss of CHO and H_2_O from the diol structure [24,25]. Based on the monoisotopic mass [M–H]–, fragmentation behavior in mass spectroscopy, and previous studies, compounds **29**–**33** were recognized as lignan molecules and previously reported in *Sargassum* sp., *Ecklonia* sp., and sea vegetables [15,16,17,19]. Among lignans, compounds **32** and **33** were tentatively reported for the first time in the SH.

Compounds **34**–**37** were previously reported in *Sargassum fusiforme, Codium* sp., and *Desmarestia antarctica*, respectively [15,19]. Sargaquinoic acid (compound **38**) and sargahydroquinoic acid (compound **39**) were previously identified in *Sargassum serratifolium*, *Sargassum fusiforme*, and *Sargassum yezoense* [15,26,27]. Compounds **40**–**42** were identified as dihydroactinidiolide, humulene epoxide II, and isoamijiol based on the mass fragmentation behavior previously reported for *Fucus virsoides* [28]. Compound **43** was identified as 3,5-dihydroxy-6,7-megastigmadien-9-one and was previously reported in the genus *Ulva*. Compounds **40** and **43** were putatively reported for the first time in the SH. Compounds **44**–**47** were identified as mojabanchromanol, fallahydroquinone, fallaquinone, and 1-H-indol-6-Carbaldehde and were previously identified in *Sargassum horneri* [29,30].

#### 3.2.3. Tannins, Sulfolipids, and Phospholipids

In SH, compounds **48**–**51** were identified as phlorotannins in various species of *Sargassum* based on the fragment behavior [31]. Compounds **52** and **53** were previously identified in the *Sargassum horneri* as 1-*O*-(11-hexadecenoly)-3-*O*-(6′-sulpho-α-D-quinovopyranosyl) glycerol and 1-*O*-hexadecanoyl-3-*O*-(6′-sulfo-α-D-quinovopyranosyl) glycerol, as they display activity in vitro and in vivo during the regulation of neuroinflammation [32]. Meanwhile, compounds **54** and **55** were identified as dioxinodehydroeckol and dibenzodioxin-1,3,6,8-tetraol, respectively, and these compounds were tentatively reported for the first time in SH.

Compounds **56**–**58** were identified as sulfolipids and phospholipids depending on their specific fragmentation patterns. These compounds have previously been reported in *Fucus vesiculosus* and *Sargassum fusiforme* (brown algae) [15,33]. Sulfolipids and phospholipids in SH are reported here for the first time.

#### 3.2.4. Carboxylic Acid, Fatty Acids, Sugars, and Other Compounds

Compounds **59**–**68** were previously reported in a variety of seaweeds and were identified as carboxylic acids based on their fragmentation patterns and previously reported in *Sargassum fusiforme*. Compounds **69**–**80** were fatty acids, compounds **81**–**83** were sugars, and compounds **84**–**89** were other compounds that were also previously reported by other seaweeds, especially *sargassum* sp. [3,15,34,35,36,37,38], as shown in Table 2.

**Table 2 antioxidants-13-00690-t002:** Putatively identified bioactive compounds in optimized *Sargassum horneri* extract by ESI-MS/MS.

Group	No	Compound Name	EF	OM *m*/*z*’	CM *m*/*z*’	MS/MS (Negative Mode)	Reference
Phenolic compounds	1.	Ferulic acid	C_10_H_10_O_4_	193.0499	193.0500	177, 149, 133	[15,16,17,19]
	2.	Sinapic acid	C_9_H_8_O_3_	163.0393	163.0395	119	
	3.	Protocatechuic acid	C_7_H_6_O_4_	153.0186	153.0186	135, 109	
	4.	Caffeic acid	C_9_H_8_O_4_	179.0344	179.0344	161, 135, 117	
	5.	Syringic acid	C_9_H_10_O_5_	197.0449	197.0450	181, 153, 125	
	6.	Salicylic acid	C_7_H_6_O_3_	137.0236	137.0244	93	
	7.	Cinnamoyl glucose	C_15_H_18_O_7_	309.0974	309.0962	147, 131, 103	
	8.	*p*-Hydroxybenzaldehyde	C_7_H_6_O_2_	121.0287	121.0289	92, 77	
	9.	Hydroxytyrosol 4-*O*-glucoside ^#^	C_14_H_20_O_8_	315.1083	315.1079	153, 123	
	10.	5-(3′,4′-Dihydroxyphenyl)-valeric acid	C_11_H_14_O_4_	209.0816	209.0813	165, 149	
	11.	5-(3′,5′-Dihydroxyphenyl)-γ-valerolactone 3-*O*-glucuronide ^#^	C_17_H_20_O_10_	383.0972	383.0978	205, 191	
	12.	Quinic acid	C_7_H_12_O_6_	191.0560	191.0555	147, 129	
	13.	Syringin	C_17_H_24_O_9_	371.1356	371.1345	353, 209, 191	
	14.	Coumaroylquinic acid ^#^	C_16_H_18_O_8_	337.0910	337.0965	293, 191, 163	
	15.	3-Sinapoylquinic acid ^#^	C_18_H_22_O_10_	397.1140	397.1171	381, 191, 129	[20]
	16.	Hydroxyferulic acid ^#^	C_10_H_10_O_5_	209.0456	209.0477	191, 177	
	17.	Caffeoyl tartaric acid ^#^	C_13_H_12_O_9_	311.0408	311.0446	293, 267, 179	
	18.	Vanillic acid 4-sulfate	C_8_H_8_O_7_S	246.9918	246.9904	203, 167.03	[21,22]
	19.	Vanillic acid	C_8_H_8_O_4_	167.0343	167.0344	151, 123.04	
	20.	3,4-Dihydroxyphenylglycol	C_8_H_10_O_4_	169.0500	169.0493	151, 123	
	21.	2-Hydroxy-2-phenylacetic acid ^#^	C_8_H_8_O_3_	151.0401	151.0498	107	
	22.	Isopropyl 3-(3,4-dihydroxyphenyl)-2-hydroxypropanoate ^#^	C_12_H_16_O_5_	239.0929	239.0925	173, 123	
Flavonoids and derivatives	23.	Epigallocatechin	C_15_H_14_O_7_	305.0652	305.0636	287, 137, 125	[15,16,23,24,25]
	24.	Epicatechin	C_15_H_14_O_6_	289.0706	289.0715	245, 109, 125	
	25.	Dalbergin	C_16_H_12_O_4_	267.0710	267.0666	251, 223, 197	
	26.	Catechin	C_15_H_14_O_6_	289.0718	289.0682	245, 205, 139	
	27.	Dihydrobiochanin A ^#^	C_16_H_14_O_5_	285.0817	285.0763	270	
	28.	Glycitein 7-*O*-glucuronide ^#^	C_22_H_20_O_11_	459.0927	459.0923	283, 267	
Lignans	29.	Arctigenin	C_21_H_24_O_6_	371.1468	371.1452	355, 327, 311	[15,16,19]
	30.	Isohydroxymatairesinol	C_20_H_21_O_7_	373.1264	373.1266	355, 343, 327, 311	
	31.	Conidendrin	C_20_H_20_O_6_	355.1174	355.1200	340, 311	
	32.	Deoxyschisandrin ^#^	C_24_H_32_O_6_	415.2127	415.2120	402, 347, 316, 301	
	33.	Secoisolariciresinol ^#^	C_20_H_26_O_6_	361.1629	361.1772	343, 331, 315	
Terpenes	34.	Carnosol	C_20_H_26_O_4_	329.1762	329.1749	285, 267	
	35.	Carnosic acid	C_20_H_28_O_4_	331.1921	331.1902	287, 269	
	36.	Loliolide	C_11_H_16_O_3_	195.1021	195.1021	161, 179, 133, 105	
	37.	Isololiolide	C_11_H_16_O_3_	195.1021	195.1013	161, 179, 133, 105	
	38.	Sargahydroquinoic acid	C_27_H_38_O_4_	425.2680	425.2691	381	[15,19,20,27,28]
	39.	Sargaquinoic acid	C_27_H_36_O_4_	423.2526	423.2579	379	
	40.	Dihydroactinidiolide ^#^	C_11_H_16_O_2_	179.1071	179.1072	135, 121	
	41.	Humulene epoxide II	C_15_H_24_O	219.1751	219.1748	201	
	42.	(−)-Isoamijiol	C_20_H_32_O_2_	303.2331	303.2319	261, 243, 225	
	43.	3,5-Dihydroxy-6,7-megastigmadien-9-one ^#^	C_13_H_20_O_3_	223.1327	223.1334	205, 187, 163	[30,31]
	44.	Mojabanchromanol	C_27_H_36_O_4_	423.2558	423.2576	379	
	45.	Fallahydroquinone	C_27_H_40_O_4_	427.2854	427.2878	383	
	46.	Fallaquinone	C_27_H_38_O_4_	425.2691	425.2740	381	
	47.	1-H-indol-6-carbaldehde	C_9_H_7_NO	144.0455	144.0441	-	
Tannins	48.	Trifuhalol-A	C_18_H_14_O_10_	389.0513	389.0496	123, 125, 139, 245, 263, 265, 353	[15,31]
	49.	Phloroglucinol	C_6_H_6_O_3_	125.0237	125.0238	97	
	50.	Fucophlorethol	C_36_H_26_O_14_	680.1154	680.1179	610,601, 495, 469, 229	
	51.	Eckol	C_18_H_12_O_9_	371.0403	371.0403	335, 317, 246, 229, 140	
	52.	1-*O*-(11-hexadecenoly)-3-*O*-(6′-sulpho-α-d-quinovopyranosyl) glycerol	C_25_H_45_O_11_S	553.2643	535.2678	317, 299, 253, 225, 207	[32]
	53.	1-*O*-Hexadecanoyl-3-*O*-(60-sulfo-α-d-quinovopyranosyl) glycerol	C_25_H_47_O_11_S	555.2824	555.2834	317, 299, 255, 224, 207	
	54.	Dioxinodehydroeckol ^#^	C_18_H_10_O_9_	369.0249	369.0284	238,195, 167, 112	
	55.	Dibenzodioxin-1,3,6,8-tetraol ^#^	C_12_H_7_O_6_	246.9919	246.9909	203, 121	[15,33]
Sulfolipids and phospholipids	56.	Sulfolipid ^#^	C_25_H_47_O_11_S	555.2848	555.2822	299, 255, 225, 206, 164	
	57.	Sulfolipid (SQMG (14:0) ^#^	C_23_H_43_O_11_S	527.2535	527.2500	225	
	58.	Phospholipid ^#^	C_25_H_44_NO_7_P	500.2733	500. 2701	303, 259, 196	
Carboxylic acids	59.	Fumaric acid	C_4_H_4_O_4_	115.005	115.0021	71	[3,15,34,35,36,37,38]
	60.	Threonic acid	C_4_H_8_O_5_	135.0290	135.0299	117, 91, 72	
	61.	Gentisic acid	C_7_H_6_O_4_	153.0187	153.0178	152, 108, 81	
	62.	Kainic acid	C_10_H_15_NO_4_	212.0922	212.0916	168, 194, 150	
	63.	Mannuronic acid	C_6_H_10_O_7_	193.0353	193.0340	175, 103, 72	
	64.	Diethyl phthalate	C_12_H_14_O_4_	221.0818	221.0843	193, 177, 149, 121	
	65.	Phthalic acid	C_8_H_6_O_4_	165.0188	165.0178	121 26, 119, 58	
	66.	3-Oxooctanoic acid	C_8_H_14_O_3_	157.0863	157.0855	139, 113, 97	
	67.	d-Glucaric acid derivative	C_12_H_14_O_10_	317.0544	317.0536	209	
	68.	Azelaic acid	C_9_H_16_O_4_	187.0969	188.0961	187, 124, 169, 111	
Fatty acids	69.	Caprylic acid	C_8_H_15_O_2_	143.1070	143.1062	125, 99, 59	
	70.	Vaccenic acid	C_18_H_34_O_2_	281.2486	281.2473	263, 223, 163, 71	
	71.	Palmitic acid	C_16_H_32_O_2_	255.2330	255.2316	237, 211, 197	
	72.	Myristic acid	C_14_H_28_O_2_	227.2015	227.2002	209, 183,179	
	73.	Octadecendioic acid	C_18_H_32_O_4_	311.223	311.2217	299, 269, 251, 223	
	74.	α-Linoleic acid	C_18_H_32_O_2_	279.2331	279.2315	261, 235, 233	
	75.	13-keto-9Z,11E-Octadecadienoic acid	C_18_H_30_O_3_	293.2125	293.2109	275, 195, 113	
	76.	10-Oxooctadecanoic acid	C_18_H_34_O_3_	297.2436	297.2420	279, 209, 141, 127	
	77.	10,16-Dihydroxy-palmitic acid	C_16_H_32_O_4_	287.2227	287.2213	269, 257, 239, 185	
	78.	2,4-Decadienal	C_10_H_16_O	151.1119	151.1113	133, 123, 119, 93	
	79.	Lauric acid	C_12_H_24_O_2_	199.1697	199.1689	181, 155	
	80.	Vernolic acid	C_18_H_32_O_3_	295.2278	295.2263	277, 251, 195, 127	
Sugars	81.	d-Galactose	C_6_H_12_O_6_	179.0568	179.0548	161, 143, 113, 101	
	82.	Mannitol	C_6_H_14_O_6_	181.0712	181.0703	165, 147, 129, 111	
	83.	Gluconic acid	C_6_H_12_O_7_	195.0517	195.0498	177, 159, 129, 98	
Other compounds	84.	3,4-Dihydroxybenzaldehyde	C_7_H_6_O_3_	137.0237	137.0244	108	
	85.	4-Hydroxyphenyl acetate	C_8_H_8_O_3_	151.0394	151.0401	106	
	86.	PGF_2α_	C_20_H_34_O_5_	353.2335	353.2320	317, 273, 235, 127	
	87.	Dinor PGF_2α_	C_18_H_29_O_5_	325.1954	325.2007	307, 199, 183, 171, 129	
	88.	Threonyl-histidylglutamic acid	C_15_H_23_N_5_O_7_	384.1518	384.1514	-	
	89.	Dihydroxyphenylalanine	C_9_H_10_O_7_NS	276.0185	276.0172	259, 231, 215, 196, 179, 150, 135	

EF: elemental formula; OM: observed mass; CM: calculated mass; (-): Negative mode; ^#^ First-time identification.

### 3.3. Attenuation of H_2_O_2_-Induced Cellular Oxidative Stress by SH

H_2_O_2_ is widely accepted as a model to induce oxidative stress and assess how that stress affects biological systems in cells and tissues. Figure 3A depicted that after treating RAW264.7 cells with H_2_O_2_, it induced cell death, which was revered in pretreated gallic acid and with SH in a dose-dependent fashion. Likewise, the SH sample also attenuated cellular ROS in a dose-dependent manner like that of gallic acid (50 μg/mL), as shown in Figure 3B. Catalase, superoxide dismutase-1 (SOD1), and other enzymes are considered as first-line in attenuating oxidative stress and in sustaining the cellular redox environment [39]. As shown in Figure 3C, treatment with H_2_O_2_ caused a significant surge in oxidative stress, which was reversed by triggering the CAT and SOD with the GA and SH pretreatment in a dose-dependent manner. 

### 3.4. Nrf2 Regulation via SH Induction of Phase II Enzymes

To regulate the phase II enzyme (HO-1), Nrf2 translocation into the nucleus is considered an important part after disintegration from the Keap1 protein. Thus, to verify whether SH can increase the phase II antioxidant enzymes through Nrf2 degradation from Keap1, we treated RAW264.7 cells with SH extract. As depicted in Figure 4A, SH dose-dependently increased the dislocation of Nrf2 from the cytoplasm to the nucleus similar to gallic acid. Because of Nrf2 translocation, the HO-1 level increases after treatment with the optimized SH extracts as illustrated in Figure 4B. This trial illustrates that SH may disrupt/degrade the Nrf2 by Keap1, resulting in the upregulation of HO-1 concentration. 

### 3.5. MAPK Activation via SH and Regulation of HO-1

The previous literature has shown that the phosphorylation of MAPKs, including JNK, ERK, and p38, can regulate the HO-1 expression in various cell types. Thus, Western blot analysis was conducted to determine the signaling pathways participating in the regulation of HO-1 expression by treating RAW264.7 cells with SH extract. As shown in Figure 5A, SH extract enhanced the phosphorylation of JNK and ERK at 45 min; however, ERK did not show any result regarding the phosphorylation. To further verify whether ERK and JNK modulate the HO-1 expression, we treated the cells with the ERK (U0123) and JNK (SP600125) inhibitor before stimulation with SH. As depicted in Figure 5B, both JNK and ERK inhibitors downregulate the HO-1 expression, which is reversed in the SH treatment. This indicates that ERK and JNK phosphorylation might be involved in the regulation of HO-1 in RAW 264.7 cells. To further verify whether either MAPK/Nrf2/HO-1 cascade has a significant role in regulating oxidative damage, pretreated with SH extracts, Raw264.7 cells were further treated with the MAPK-specific (U0123; SP600125) inhibitors to analyze the production of ROS. As shown in Figure 5C, H_2_O_2_ induced the production of ROS, which was downregulated via the SH extract. However, when treated with specific inhibitors, the trend was again reversed, which reflects the involvement of MAPK pathways in oxidative stress. 

### 3.6. Docking Results

Numerous studies have revealed that molecular docking is an extremely useful tool for studying ligand–protein interactions during receptor activation or inhibition to better understand the mechanism of action [18]. Based on previous metabolite profiling and Western blot results, we hypothesized that tentatively identified compounds have the ability to attenuate oxidative stress. In order to prove this hypothesis, we performed a molecular docking simulation to identify compounds and calculate the binding affinity of bioactive compounds with Keap1 protein (4L7B) using AutoDoc Vina [40] and MolDock [41]. Fifty-five compounds were checked for the binding energy calculation (Appendix A), and those bioactive compounds that showed more binding energy (cutoff value ≤ −9) in AutoDoc Vina were further validated and compared in MolDock. Among them, five compounds exhibited the highest energy, and sargaquinoic acid glycitein 7-*O*-glucuronide (Table 3) exhibited the highest binding affinity in AutoDoc Vina (−9.20 Kcal/mol; 9.52 Kcal/mol) and MolDock (−155.842; −148.975), respectively.

## 4. Discussion

An unbalanced and uncontrolled production of RNS and ROS results in oxidative stress/damage progression in a cellular environment by reacting with proteins, lipids, and DNA [3]. In order to attenuate the uncontrolled ROS/RNS generation, phytochemicals with intrinsic antioxidants play a significant role by regulating the cellular protective signaling cascades [5]. Numerous trials have been conducted in order to find the appropriate bioactive agent; however, marine seaweeds, especially marine brown algae, are still underutilized plant resources. *Sargassum horneri* (SH) is widely used in the Asia and Pacific region due to its biological and pharmacological properties, as this species contains various bioactive compounds [14]. The literature showed that few studies have been conducted to isolate some certain bioactive compounds ((-)-lolidolide, catechin, epicatechin, etc.), which exhibited anti-inflammatory, anti-oxidative activities [14,15,16]. Although, until today, no study has been conducted to optimize the multifrequency ultrasonic extraction conditions to achieve higher antioxidant activity followed by the identification of bioactive compounds using HRMS in the optimized extract or further check the underlying mechanism of action using in vitro and in silico assays to attenuate the oxidative stress. Keeping these limitations in view, the current trial was conducted to achieve the highest antioxidant activity using sophisticated statistical techniques (RSM and ANNs) to optimize the multifrequency ultrasonic extraction and identification of bioactive compounds in the optimized extracts. Furthermore, the attenuation of oxidative stress was assessed via Nrf2/MAPKs/HO-1 cascade followed by checking the binding energies of identified compounds with Keap1 protein (4L7B) using molecular docking software (AutoDoc and MolDock). 

To optimize the extraction conditions, the RSM is frequently used as an appropriate statistical technique for modeling complicated multivariate processes wherein responses are affected by several factors, such as time, temperature, equipment intensity, and power [16]. The statistical significance of the fitted RSM model can be calculated using the ANOVA results, which also contained other important model parameters, such as the regression coefficient (β), coefficient of variation (CV), adjusted correlation factor (R^2^), F-value, *p*-value, adequate precision, model significance, and lack of fit. In the current trial, in the RSM model, for TPC extraction, the concentration (X_1_); time (X_2_); temperature (X_3_); the interactions of time (X_2_) × temperature (X_3_) and temperature (X_3_) × frequency (X_4_); and the squares of temperature (X_3_^2^), time (X_2_^2^), concentration (X_1_^2^), and frequency (X_4_^2^) exhibited a significant effect. In the case of the TFC response, the concentration (X_1_) and the squares of concentration (X_1_^2^) and temperature (X_2_^2^) were significantly affected by the TFC extraction. For DPPH, the concentration (X_1_); time (X_2_); the interactions of concentration (X_1_) × frequency (X_4_) and time (X_2_) × temperature (X_3_); and the squares of concentration (X_1_^2^) and time (X_2_^2^) exhibited a significant effect, whereas, in the ABTS response, the concentration (X_1_) and squares of concentration (X_1_^2^), time (X_2_), temperature (X_3_), and frequency (X_4_^2^) also exhibited a significant effect on the ABTS content. Similarly, the CV value, coefficient of variation (CV), and adjusted correlation factor (R^2^) are also in the range (Appendix A), as expected, which was further verified via the previous results of Alshammari et al. [40]; they optimized the heat reflex extraction conditions of Ajwa dates and calculated the R^2^, F-values, *p*-values, and, most importantly, the model significance criteria and lack of fit as ‘non-significant’. Recently, Javed et al. [16] optimized the microwave-assisted extraction conditions of *Sargassum fusiforme* (a subspecies of *Sargassum*), which supports our results; however, the MUAE approach reflects the better extraction conditions by keeping the extraction temperature lower. Similarly, Choi et al. [19] determined the optimal heat reflux extraction conditions for the recovery of bioactive ingredients from *Nypa fruticans* Wurmb. using RSM and ANN models, which also supports the current results. However, the current trial is better in terms of the optimizing factors and extraction method because Choi et al. [17] used three factors (time, temperature, and ethanolic concentration) and used a conventional method to extract the bioactive compounds. 

ANNs have also emerged as practical and powerful nonlinear computational tools for complicated nonlinear processes owing to their superior learning and predictive modeling capabilities [15]. Principally, ANNs are based on the human central nervous system (CNS), where a complex network of interconnected neurons is capable of computing in response to input data [14]. The ANN model was trained at a topology of 4–10–4 and showed better validation performance and less histogram error as depicted in Figure 2. After optimizing the RSM and ANN models, the prediction abilities of both models were determined and compared using numerous statistical metrics, such as the absolute average deviation (AAD), root mean square error (RMSE), R^2^ (coefficient of determination), and standard error of prediction (SEP) [40,41]. The ANN model prediction abilities are better than that of RSM as exhibited in Appendix A. The present outcomes are consistent with the results of Choi et al. [19] as they also optimized the extraction condition for *Nypa fruticans* Wurmb and reported that the ANN model is better in terms of prediction. Alshammari et al. [40] also reported on the accuracy of the ANN model in terms of prediction in comparison with RSM prediction, which is also consistent with our present results. Although both studies predicted the values for only three independent factors (time, temperature, and solvent concentration), in the current study, four independent factors were used.

Hydrogen peroxide (H_2_O_2_) is widely accepted as a model to induce cellular oxidative to investigate to assess how oxidative stress affects biological systems in both cells and tissues [42]. The current trial depicted that optimized extract dose-dependently attenuates ROS generation by regulating the CAT and SOD; see Figure 3A–C. The present outcomes showed that SH helps in promoting the regulation of enzyme proteins, which in turn attenuate oxidative damage by maintaining the cellular redox balance. The literature has shown that plants/food enriched with bioactive compounds, especially polyphenolic compounds, enhance the SOD1 and CAT levels, which results in mitigating oxidative stress. In the previous study, we also optimized the *Sargassum fusiforme* ethanolic extract and determined its activity against oxidative stress, which downregulates oxidative stress by enhancing the SOD and CAT enzymes. Likewise, various investigations also illustrate that the administration of brown algae, especially *Sargassum* sp., helps in reducing oxidative damage and stress. Numerous studies have exhibited that polyphenolics, such as protocatechuic acid, vanillic acid, gallic acid, catechin, and epicatechin (tentatively detected in the SH sample), can upregulate the endogenous antioxidant system, leading to cellular protection from oxidative stress [3,5,15,16]. Consequently, it is assumed that the surge in the first-line antioxidant enzymes via SH might be due to the presence of phenolics and flavonoids, which could be present in higher concentrations during the optimization in the current study. The literature has shown that the presence of bioactive compounds, especially gallic acid and loliolide, scavenged the free radical chain reaction by transferring hydrogen atoms [43]. Likewise, ethanolic, methanolic, and acetone extracts, etc., and extracted polysaccharides were used to check the antioxidant activity or underlying mechanism of action without quantifying the bioactive compounds, but the current trial tries to show the possible compounds that could be behind attenuating the activity against diseases. 

The literature has exhibited that phase II antioxidant enzymes, especially heme oxygenase-1 (HO-1), impart an important role in downregulating the ROS and are regulated by the Nrf2, a central controller of ARE-driven antioxidant gene expression. In a normal state, Nrf2 activity is strictly managed by the Kelch-like ECH-associated protein 1 (Keap1) as an adaptor protein for Cullin-3 (Cul3)-dependent E3 ubiquitin ligase enzyme, which manages the Nrf2 degradation and ubiquitination [3,5]. Optimized SH extract helps in the translocation of Nrf2 into the nucleus, which results in the activation of HO-1 (phase II enzyme protein) in a dose-dependent manner (Figure 4A,B). Numerous previous studies have indicated that *Sargassum* sp., including *Sargassum fusiforme* and *Sargassum serratifolium*, actively upregulate HO-1 expression through the regulation of Nrf2 [44]. Likewise, in our previous study, we conducted the metabolite profiling of *Sargassum fusiforme,* which exhibited the presence of various polyphenolic compounds including protocatechuic acid, vanillic acid, gallic acid, and naringenin, which were also present in SH extract because of same *Sargassum* genus (Table 2). Thus, processing conditions might alter the concentrations of the bioactive compounds that need to be carefully selected during the compound’s extraction or separation.

Previously, studies have exhibited that MAPK phosphorylation could be involved in the regulation of phase II enzyme HO-1. SH extract upregulates the HO-1 levels dose-dependently, which was further verified by treating the RAW264.7 cells with the specific inhibitors ERK (U0123) and JNK (SP600125) (Figure 5A–C). Previous studies have illustrated that the presence of certain bioactive compounds, including fucosterol, loliolide, sargachromenol, gallic acid, 3-Hydroxy-5,6-epoxy-β-ionone, etc., detected or isolated from SH resulted in the attenuation of oxidative stress and inflammation [14,15,45]. Furthermore, Alam et al. [5] checked the activity of *Nymphaea nouchali* (Burm. f) stem extracts along with metabolite profiling using a high-resolution mass spectrometer and reported different bioactive compounds, including gallic acid, naringin, epicatechin, catechin, 5-*O*-caffeoylquinic acid, 3-Feruloylquinic, etc., resulting in the attenuation of oxidative stress through the regulation of MAPK/NRF2/HO-1/ROS signaling. In the current trial, we use a novel approach to extract the bioactive compounds, including lignans, terpenes, tannins, etc., which have not been previously reported together in a trial [3,5], which could be the reason for the stronger antioxidative stress activity exhibited by SH. 

Today, in silico approaches are attracting more attention as these modern computational and experimental methodologies may be combined more effectively than a battery of laboratory experimental analysis [46,47]. For this purpose, MolDoc and AutoDoc molecular docking simulations were carried out on the 55 selected and identified compounds against the Keap1 protein (4L7B). Among them, five compounds exhibited the highest energy, and sargaquinoic acid glycitein 7-*O*-glucuronide (Table 3) exhibited the highest binding affinity in AutoDoc Vina (−9.20 Kcal/mol; 9.52 Kcal/mol) and MolDoc (−155.842; −148.975), respectively. According to the literature, one Nrf2 molecule is attached to a Keap1 dimer through two distinct motifs (DLG and ETGE) in the neh2 region via a “hinge and latch” mechanism. The ETGE and DLG motifs form a β-turn structure through the electrostatic interaction among glutamate and acidic aspartate with Arg380, Arg415, and Arg483 in the Kelch region of Keap1 [48]. Therefore, these three amino acids gained more importance during the ligand–protein interaction. Li et al. [30] used the egg-derived tri-peptide proteins to inhibit the direct interaction of Keap1 with Nrf2 using molecular docking, and the results showed that DKK and DDW proteins bind with Arg380, Asn382, Arg415, Arg482, and Ser508, respectively. Likewise, Adelusi et al. [48] determined the quantum mechanics, dynamics, and docking experiments to identify the Keap1 inhibitors, and among 50 antioxidant compounds, maslinic acid exhibited the highest binding energy (−10.6 kJ/mol). Among different amino acid residue interactions, maslinic acid forms van der Waals interactions with Arg415 of the Keap1 protein. Recently, Diniyah et al. [17] also determined the binding ability of four bioactive compounds, i.e., gallic acid, coumaric acid, epicatechin, and catechin, for Keap1 protein, and the results showed that these four compounds also form bonds with different amino acids, especially Arg415. In the current experiment, sargaquinoic acid forms one hydrogen interaction with Arg483, whereas glycitein 7-*O*-glucuronide forms a pi–pi interaction with Arg415 (Figure 6A–D), which is parallel with the previous studies [17]. 

## 5. Conclusions

After optimizing the extraction condition, 89 bioactive compounds were reported in optimized SH extract, and among them, 19 compounds, including phenolic acid, flavonoids, terpenes, lignan, etc., were tentatively reported for the first time in SH. The in vitro results indicated that optimized SH extract attenuates the oxidative stress in a dose-dependent fashion via the Nrf2/MAPK/HO-1 signaling pathway due to the presence of active ingredients, including gallic acid, epicatechin, catechin, 5-*O*-caffeoylquinic acid, and 3-feruloylquinic, etc. For further validation, in silico simulation was conducted on the Keap1 protein using 55 bioactive compounds and the result indicated that sargaquinoic acid and glycitein 7-*O*-glucuronide could potentially bind with Keap1, as exhibited by higher binding energy (9.20 Kcal/mol; 9.52 Kcal/mol), by forming a hydrogen bond with Arg483 and Arg415. The current investigation provides an alternative statistical technique and supports a preferred extraction technology for identifying important bioactive compounds that could be otherwise present in lower concentrations under non-optimized conditions. Because temperature, time, and solvent concentration greatly affect extraction yield, multivariant optimization is a promising approach instead of single-factor optimization. This trial provides the basis for the scientific community to isolate/extract compounds that could be used in broad commercial applications as promising ingredients for the development of functional foods and nutraceuticals. However, further in vitro and in vivo research should be conducted to check the toxicity and mechanism of action of identified compounds, especially sargaquinoic acid and glycitein 7-*O*-glucuronide, against oxidative stress and related diseases. Furthermore, the dynamics and kinetics of these compounds should also be checked using YASARA to visualize protein interaction to validate our in silico results.

## Figures and Tables

**Figure 1 antioxidants-13-00690-f001:**
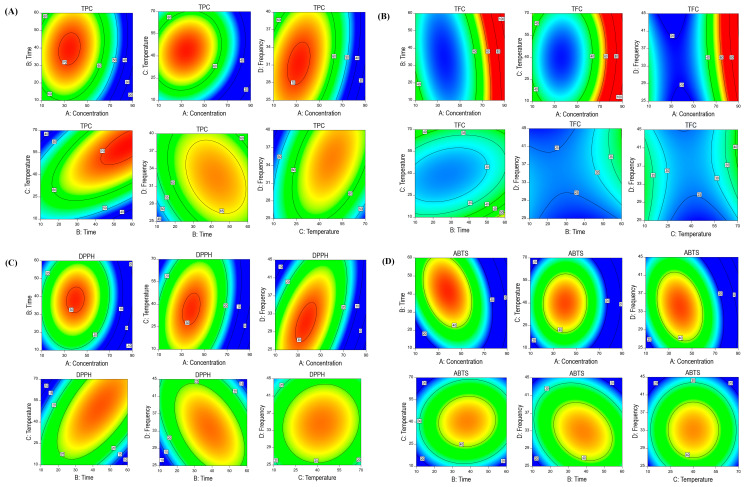
RSM interaction plots of SH extraction on ethanol concentration, time, temperature, and ultrasound frequency using BBD design for TPC (**A**), TFC (**B**), DPPH (**C**), and ABTS (**D**).

**Figure 2 antioxidants-13-00690-f002:**
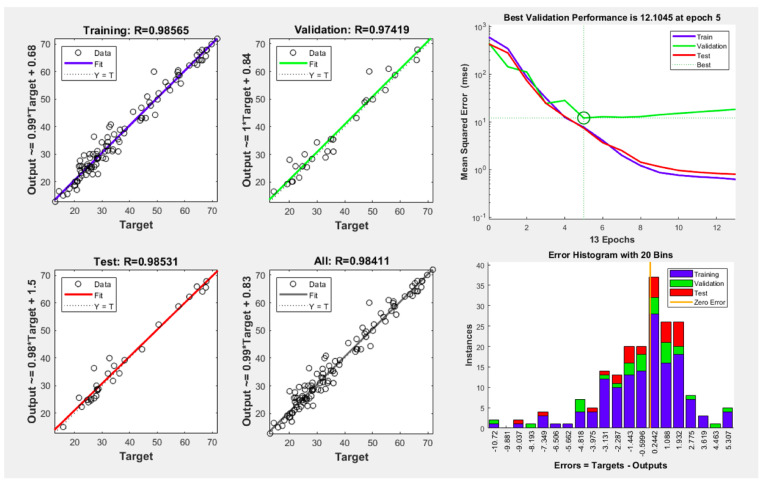
Optimal topology and trained model performance curves with epoch numbers for the dependent variables of the developed ANN model.

**Figure 3 antioxidants-13-00690-f003:**
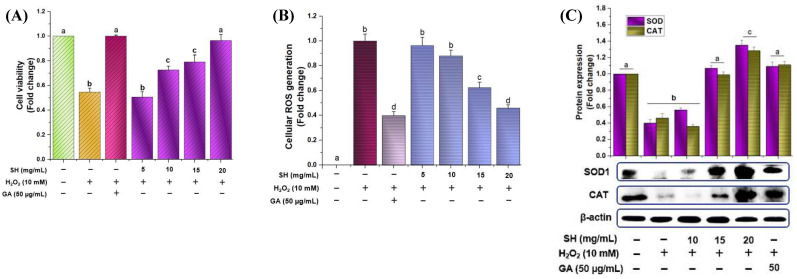
Effect of optimized SH extract on cell viability, ROS generation, and primary enzymes treated with and without H_2_O_2_ in RAW264.7 cells. Cells were pretreated with SH and gallic acid for 12 h, followed by treatment with or without 10 mM of H_2_O_2_ for 6 h. Cell viability with H_2_O_2_ (**A**). The generation of cellular ROS was evaluated using the DCFH-DA method (**B**); the protein expression of SOD1 and catalase was measured with Western blot analysis (**C**), and the relative protein expression was quantified using Image J software (Version 1.54i 03). Differences in the alphabetic letters represent statistical significance (*p* < 0.05) to one another.

**Figure 4 antioxidants-13-00690-f004:**
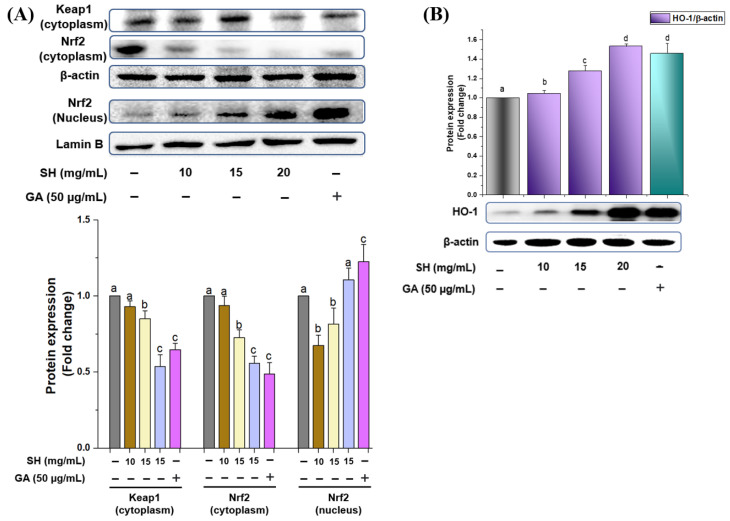
Activation of phase II antioxidant enzymes via the regulation of Nrf2. Optimized SH extract dose-dependently decreases the Keap1 and facilitates the translocation of Nrf2 protein expression into the nucleus (**A**), resulting in the upregulation of phase II enzyme expression (**B**) using immunoblotting assay. The relative protein expression was quantified using Image J software. Differences in the alphabetic letters represent statistical significance (*p* < 0.05) to one another. Gray: non treated; purple: sample treated; green: gallic acid treated.

**Figure 5 antioxidants-13-00690-f005:**
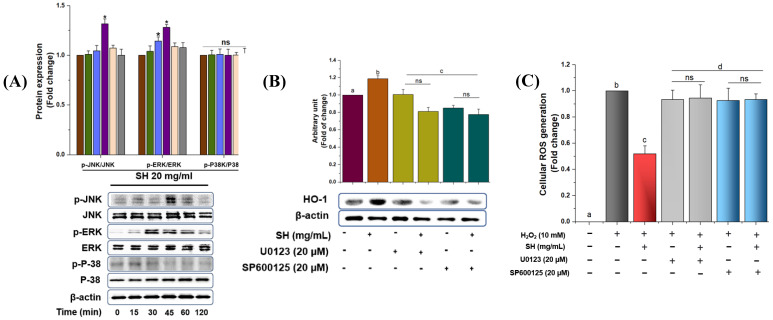
Activation of ERK, p38, and JNK via SH results in Nrf2 translocation. RAW 264.7 cells were treated with SH (20 mg/mL) at various times, and kinase activity was determined with immunoblot assay (**A**). Cells were treated with optimized SH extract and specific inhibitors, SP600125 (JNK inhibitor) and U0123 (ERK), for 1 h, and HO-1 protein levels were analyzed via Western blot analysis. Red: 0 min; green: 15 min; blue: 30 min; purple: 45 min, paste: 60 min, and gray: 120 min (**B**); the production of ROS; red: non-treated; orange: sample; yellow: sample with ERK inhibitor, green: sample with JNK inhibitor. (**C**) and the relative protein expression were quantified using Image J software gray: model control, red: SH; light gray: SH+ERK inhibitor, and sky:SH+JNK inhibitor. Differences in the alphabetic letters represent statistical significance (*p* < 0.05) to one another.

**Figure 6 antioxidants-13-00690-f006:**
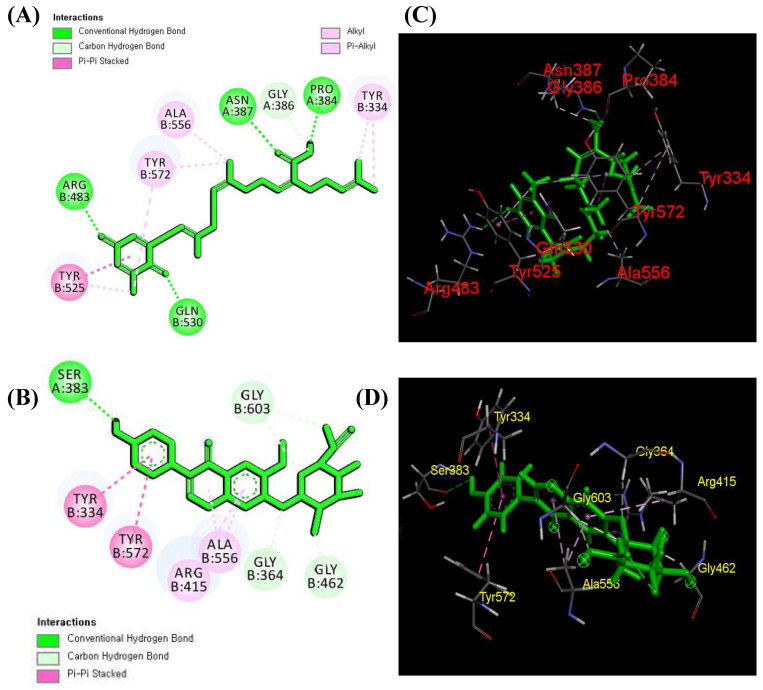
Molecular interaction results of sargaquinoic acid with Keap1 protein (4L7B). The 2D molecular interaction of sargaquinoic acid (**A**) and of glycitein 7-*O*-glucuronide (**B**) and the 3D molecular interaction of sargaquinoic acid (**C**) and of glycitein 7-*O*-glucuronide (**D**) with surrounding Keap1 protein amino acids.

**Table 3 antioxidants-13-00690-t003:** AutoDoc Vina and MolDock comparative binding energy scores for selected compounds.

Identified Compounds	MolDock Score	Chimera (Kcal/mol)
Glycitein 7-*O*-glucuronide	−148.975	−9.5
Sargaquinoic acid	−155.842	−9.2
Carnosol	−108.911	−9.0
Conidendrin	−149.703	−9.0
Eckol	−140.451	−9.1
Isohydroxymatairesinol	−150.118	−9.1

## Data Availability

The data will be available upon request.

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
