# Peer review of "RSM- and ANN-Based Multifrequency Ultrasonic Extraction of Polyphenol-Rich Sargassum horneri Extracts Exerting Antioxidative Activity via the Regulation of MAPK/Nrf2/HO-1 Machinery"

_antioxidants, 2024, doi:10.3390/antiox13060690_

Round 1

Reviewer 1 Report

The article ‘RSM and ANN-based multifrequency ultrasonic extraction of 2 polyphenol-rich Sargassum horneri extracts exerting antioxidative activity via the regulation of MAPK/Nrf2/HO-1 machinery’ is an interesting paper that presents the results on the biological potential of different extracts from S. hornei. This research presents the in vitro and in silico analysis, which performs important data in the area of antioxidant activity. Moreover, the article presents an interesting approach to the optimization of extraction, and the profile of the bioactive compounds responsible for those biological activities was evaluated using HRMS. I find this article valuable. Despite this, I suggest some minor changes, which I present below

Line 70-71: „So far, various marine bioactive ingredients have been isolated from SH and studied against different diseases such as  inflammation and oxidative stress.” - inflammation and oxidative stress are not diseases

In some places, in vitro, in vivo, or silico are written italics, and in some, without

Sargassum horneri should be written in italics (the title of Table 2)

(…) SH (10 g) was soaked for one hour before extraction with 150 ml of solvent (…) – which solvent was used to prepare the extract? It should be added to the methods.

Some basic information should be added to point 2.3 – the concentration of examined extracts, the names of references used, the mode of expression of the results, the quantities of experiment repetitions, and thus, the specific features that can be unique for this experiment.

Is the title of Figure 3. ‘Effect of optimized SF extract on cell viability…’ is correct?

Response surface methodology 486 and 61 – the abbreviation should be introduced only once

A more precise caption of Figure 1 is needed (as the title of the figure)

Table 2 has a caption: ‘ Putatively identified bioactive compounds in optimized Sargassum horneri extracts by ESI-MS/MS.’ Which extract is presented in this table?

The authors indicated that some of the extracts were more active (DPPH, ABTS) or contained more compounds (TPC, TFC). Which of the following were studied using cellular biological analysis? 

In Figure 5, there is „SH (mg/ml),” but in the caption of the figure, it is SH (20 µg/mL). What is correct?

The conclusion should be modified – it is the summary rather (very similar to the abstract), and perhaps a more consistent conclusion would be better.

Author Response

Reviewer 1

The article ‘RSM and ANN-based multifrequency ultrasonic extraction of 2 polyphenol-rich Sargassum horneri extracts exerting antioxidative activity via the regulation of MAPK/Nrf2/HO-1 machinery’ is an interesting paper that presents the results on the biological potential of different extracts from S. hornei. This research presents the in vitro and in silico analysis, which performs important data in the area of antioxidant activity. Moreover, the article presents an interesting approach to the optimization of extraction, and the profile of the bioactive compounds responsible for those biological activities was evaluated using HRMS. I find this article valuable. Despite this, I suggest some minor changes, which I present below.

  1. Line 70-71: So far, various marine bioactive ingredients have been isolated from SH and studied against different diseases such as inflammation and oxidative stress.” - Inflammation and oxidative stress are not diseases

Response: Thank you for your insightful comments. We apologize for the negligence; we replaced the text as the reviewer suggested (lines 63–64).

  1. In some places, in vitro, in vivo, or silico are written italics, and in some, without.

Response: Thank you for your valuable comments. We apologized again for our mistake; we carefully rechecked the whole text and fixed the mistake where required.

  1. Sargassum horneri should be written in italics (the title of Table 2).

Response: Thank you for your meaningful comment. We rechecked the whole text and replaced the sargassum horneri with italics where required.

  1. (…) SH (10 g) was soaked for one hour before extraction with 150 ml of solvent (…) – which solvent was used to prepare the extract? It should be added to the methods.

Response: Thank you for your kind suggestion. We have mentioned that the SH samples were dissolved in different concentrations of ethanol as per the Box-Behnken design generated by the RSM (Table 1).

  1. Some basic information should be added to point 2.3 – the concentration of examined extracts, the names of references used, the mode of expression of the results, the quantities of experiment repetitions, and thus, the specific features that can be unique for this experiment.

Response: Thank you for your important and valuable comments. We have updated the text by adding the necessary information required for easy understanding (lines 123-131).

  1. Is the title of Figure 3. ‘Effect of optimized SF extract on cell viability…’ is correct?

Response: Thank you for your important observation. We apologize for the mistake, and now we have corrected it and rechecked the whole text to fix this mistake.

  1. Response surface methodology 486 and 61 – the abbreviation should be introduced only once.

Response: We have updated the text as per what the reviewer suggested.

  1. A more precise caption of Figure 1 is needed (as the title of the figure).

Response: We have changed the title of the figure according to the reviewer’s suggestion.

  1. Table 2 has a caption: ‘Putatively identified bioactive compounds in optimized Sargassum horneri extracts by ESI-MS/MS.’ Which extract is presented in this table?

Response: Thank you for your valuable observation. Actually, we have identified the bioactive compounds after optimizing the condition, which are TPC (66.34 mg GAE/g), TFC (43.63 mg CAT/g), DPPH (29.11% inhibition), and ABTS (46.77% inhibition), respectively (lines 249-251).

  1. The authors indicated that some of the extracts were more active (DPPH, ABTS) or contained more compounds (TPC, TFC). Which of the following were studied using cellular biological analysis?

Response: Thank you for your important and valuable comments. As in the previous comment, we have mentioned that we used the optimized SH extract (lines 249-251) for identifying the bioactive compounds and also for the biological activity.

  1. In Figure 5, there is “SH (mg/ml),” but in the caption of the figure, it is SH (20 µg/mL). What is correct?

Response: Thank you for your kind suggestion. We apologize for our negligence; actually, it is 20 mg/ml and we have updated the text as per the reviewer’s suggestion.

  1. The conclusion should be modified – it is the summary rather (very similar to the abstract), and perhaps a more consistent conclusion would be better.

Response: Thank you for your meaningful comment. We have revised the summary as per the reviewer’s suggestion.

Reviewer 2 Report

The manuscript submitted by Javed et al. represents an innovative approach for the optimization of ethanolic extractions of Sargassum horneri and investigation of the mechanism of its antioxidant activities in vitro. Based on these data, there is some advancement in the understanding of the mechanism of action of Sargassum extracts compared with previous works such as https://doi.org/10.3390/antiox10060859 and https://doi.org/10.1186/s12906-018-2314-6. Moreover, the extraction approach is advanced.

However, there are some things that need to be improved. Some observations were given in the other sections.

Here I will emphasize only on the Introduction. Although it presents the relevant aspects, it should be improved, because the jump from one subject to another is not smoothly done and it seems like there are several unrelated narratives.

For example, at line 59 the authors suddenly start to talk about extraction methodologies, without any relation to the biochemical systems described before. I propose the Introduction to be rearranged a little. After the description of the proteins involved (after line 59) maybe the Information about Sargassum would fit better (lines 67-76, because it was already mentioned at lines 45-47 about phytochemicals), followed by the text about ANN and RSM (lines 59-66) and next lines 76-85.

Please define BBD first time it is mentioned.

Line 263 – please correct “more better” with only “better”.

Table 1 does not seem to be cited in the text. It seems that at lines 318 and 367 it should be Table 2.
The Table with the docking results should be Table 3 and same in the text.

Lines 576-577 – not sure from where the conclusion with the donation of hydrogen atoms.

Author Response

Reviewer: 2

The manuscript submitted by Javed et al. represents an innovative approach for the optimization of ethanolic extractions of Sargassum horneri and investigation of the mechanism of its antioxidant activities in vitro. Based on these data, there is some advancement in the understanding of the mechanism of action of Sargassum extracts compared with previous works such as https://doi.org/10.3390/antiox10060859 and https://doi.org/10.1186/s12906-018-2314-6. Moreover, the extraction approach is advanced.

However, there are some things that need to be improved. Some observations were given in the other sections.

  1. Here I will emphasize only on the Introduction. Although it presents the relevant aspects, it should be improved, because the jump from one subject to another is not smoothly done and it seems like there are several unrelated narratives.

For example, at line 59 the authors suddenly start to talk about extraction methodologies, without any relation to the biochemical systems described before. I propose the Introduction to be rearranged a little. After the description of the proteins involved (after line 59) maybe the Information about Sargassum would fit better (lines 67-76, because it was already mentioned at lines 45-47 about phytochemicals), followed by the text about ANN and RSM (lines 59-66) and next lines 76-85.

Response: Thank you for your valuable and kind suggestion. We have rearranged the text according to the reviewer’s comment.

  1. Please define BBD first time it is mentioned.

Response: Thank you for your comment. We have updated the text as per the reviewer’s suggestion.

  1. Line 263 – please correct “more better” with only “better”.

Response: Thank you for your suggestion. We have corrected the text.

  1. Table 1 does not seem to be cited in the text. It seems that at lines 318 and 367 it should be Table 2. The Table with the docking results should be Table 3 and same in the text.

Response: Thank you for pointing out the mistake. We apologize for this error. Now we have rechecked the whole text and made the necessary corrections where needed.

  1. Lines 576-577 – not sure from where the conclusion with the donation of hydrogen atoms.

Response: Thank you for your important observation. We have removed the unwanted lines from the text and also revised the whole text to fix such errors.

Reviewer 3 Report

In general, the manuscript antioxidants-2979925 is interesting, extensive, well-planned and executed.

It seems to me that the text needs linguistic correction, there are quite a few errors.

Please see attached pdf document, it contains all specific comments to the paper.

Author Response

Reviewer: 3

In general, the manuscript antioxidants-2979925 is interesting, extensive, well-planned and executed.

It seems to me that the text needs linguistic correction, there are quite a few errors.

  1. The term ’new compounds’ is rather reserved for metabolites whose chemical structures are reported for 1st time (elucidation performed by NMR etc).

Response: Thank you for the meaningful comments. Yes, we agreed with the reviewer’s comment. These compounds have already been identified or reported in other marine seaweeds, but in the SH, they are putatively or tentatively identified or reported for the first time, and we have also updated the text accordingly. However, further verification and validation are required using the compounds standard, which might be difficult to separate or unavailable in pure form.  

  1. Please replace variables with the response.

Response: Thank you for your observation. We have updated the text as per the reviewer’s suggestion.

  1. Why did the phytochemical analysis use direct injection on MS and not use UHPLC-MS? Chromatographic separation of complex mixtures (extracts) greatly facilitates the identification of metabolites and significantly increases its accuracy and confidence.

Response: Thank you for your important and valuable comment. We agree with the reviewer’s suggestion that UHPLC-MS greatly facilitates compound identification, although we used the direct injection method due to time efficiency, simplicity, and resource optimization, whereas in UHPLC, which is a more complex method as it takes too much time to optimize the conditions, i.e., to set the gradient, many experiments need to be run using the hit-and-trial methods. Moreover, different columns are needed to identify different compounds, which also increases the cost. In summary, our choice of direct injection MS was a deliberate decision based on the balance between time, efficiency, simplicity, and resource optimization. We acknowledge the trade-offs associated with this approach and remain open to further discussion and exploration of alternative methodologies to enhance the robustness of our analyses. 

  1. SM file not included. As a reviewer, I have no way to verify the information given in the SM and referred to in the text when it is unavailable to me.

Response: Thank you for reporting our negligence, and we apologize for this mistake. Now, we have provided the supplementary data with this revision. 

  1. Rewrite “ Flavonoids are the primary class of the phenolic acids”

Response: Thank you for pointing out the mistake. We have rewritten the line.

  1. Compd 42 was identified as isoamijiol (Table 2).

Response: Thank you for your important comment and for also highlighting our mistake. We have made the necessary corrections to the text. 

  1. check correctness

Response: Thank you for your kind suggestion. We have corrected the mistake.

  1. Consider adding column 'Reference' at the end of table, and list there all papers reporting the occurrence of metabolite in Sargassum genus. This can shorten and make it easier to describe the results in Section 3.2.

Response: Thank you for your valuable comment. We have added the reference at the end of Table 2 according to the reviewer’s suggestion.

  1. Sargagassum horneri italics

Response: We have italics the Sargassum horneri in the whole manuscript. 

  1. I do not understand, consider changing to [M-H]-, m/z

Response: We have updated the text according to the reviewer’s suggestion.

  1. Hypnotized and preformed?

Response: We have rechecked the whole manuscript and fixed these kinds of mistakes.

  1. Dependent should be changes to independent

Response: Thank you for your suggestion.

Reviewer 4 Report

The manuscript reports on the study of Sargassum horneri extraction of metabolites and the antioxidant effect on murine macrophage cell line, RAW 264.7.

My impression is that this manuscript is just a collection of three different short communications just hastily put together for this submission, without carefully checking the text.

My suggestion to the Editor is to reject the manuscript and ask the authors to deeply review and proofread their work.

Just few examples

In line 641 is clearly written  =>  Supplementary Materials: Not applicable. But in lines 194, 201 and 493 the Authors refer to Table 1S, but of course no Table 1S is present in the manuscript, such as Table 2S and 3S.

Line 271 Table 1. Comparison of Box-Behnken design (BBD) for independent variables against corresponding target responses (experi-271 mental), RSM and ANN predicted values. But at line 316 is stated < Likewise, glycitein 7-O-glucuronide (28) were re-316 ported in Centroceras sp., but both compounds were putatively identified in SH for the first time in this study (Table 1). In addition, is clearly written, line 365 < Compounds 69–80 were fatty acids, compounds 81–83 were sugars, 365 compounds 84-89 were other compounds which were also previously reported by other 366 seaweeds especially sargassum sp. [3, 17, 34-38], as shown in Table 1.>. Table 1 is becoming ubiquitous.

Same story for Table 2, see lines 369 and 453.

RSM and ANN results are useless without the proper statistics which are missing.

Docking is also useless without knowing the detail of the analysis, e.g. ionic strength, water, pH, metals, forcefield and so on.

The text and the graphics must be improved.

These are just a few examples as it is impossible to give an objective judgement.

See the major comments

Author Response

Reviewer: 4

The manuscript reports on the study of Sargassum horneri extraction of metabolites and the antioxidant effect on murine macrophage cell line, RAW 264.7.

My impression is that this manuscript is just a collection of three different short communications just hastily put together for this submission, without carefully checking the text.

My suggestion to the Editor is to reject the manuscript and ask the authors to deeply review and proofread their work.

Just few examples

  1. In line 641 is clearly written  =>  Supplementary Materials: Not applicable. But in lines 194, 201 and 493 the Authors refer to Table 1S, but of course no Table 1S is present in the manuscript, such as Table 2S and 3S.

Response: Thank you for your critical observation and valuable comment. We apologize for this, actually during the paper submission, we added the supplementary file, but mistakenly it was not uploaded with the manuscript. Now, the supplementary file is uploaded with the updated manuscript version.

  1. Line 271 Table 1. Comparison of Box-Behnken design (BBD) for independent variables against corresponding target responses (experi-271 mental), RSM and ANN predicted values. But at line 316 is stated < Likewise, glycitein 7-O-glucuronide (28) were re-316 ported in Centroceras sp., but both compounds were putatively identified in SH for the first time in this study (Table 1).In addition, is clearly written, line 365 < Compounds 69–80 were fatty acids, compounds 81–83 were sugars, 365 compounds 84-89 were other compounds which were also previously reported by other 366 seaweeds especially sargassum sp. [3, 17, 34-38], as shown in Table 1.>. Table 1 is becoming ubiquitous.

Response: Thank you for pointing out the mistake. We have rechecked the whole text and corrected the mistake.

  1. Same story for Table 2, see lines 369 and 453.

Response: Thank you for your kind suggestion. We have rechecked the text and fixed the mistake.

  1. RSM and ANN results are useless without the proper statistics which are missing.

Response: Thank you for your important observation. Actually, we have provided the RSM and ANN statistical analyses in the supplementary file, which was mistakenly not uploaded during the manuscript submission. We apologize for this mistake, and now the file is uploaded with the revised version of the paper.

  1. Docking is also useless without knowing the detail of the analysis, e.g. ionic strength, water, pH, metals, forcefield and so on.

Response: Thank you for your valuable suggestion. During molecular docking, we usually consider the active pocket site of the protein, binding energy between the ligand and the protein, and the type of interaction (hydrogen bonding, pi-pi interaction, etc.) between the ligand and surrounding amino acids, etc. (please refer to lines 558–562). Before conducting the analysis, we also prepare the ligand and protein by removing the unnecessary moieties, i.e., water molecules. Moreover, it also depends from software to software. Please refer to the following paper: https://doi.org/10.3390/molecules200713384. For calculating the water, pH, and force field, usually SAR analysis and YASARA software might be used, but in our study, we did not use those software devices.

  1. The text and the graphics must be improved.

Response: Thank you for your kind suggestion. We have now updated the whole text and graphics of the paper according to the reviewer’s suggestion.

Round 2

Reviewer 3 Report

The 2nd version of manuscript antioxidants-2979925 is improved and SM file is now attached.

I currently have a few comments that are related to the information in SM (which was previously missing) and other. All comments are included in the attached pdf document.

Author Response

May 9, 2024

Prof. Alessandra Napolitano

Editors-in-Chief

Antioxidants

Dear Editor-in-Chief:

Please find enclosed our revised manuscript titled “RSM and ANN-based multifrequency ultrasonic extraction of polyphenol-rich Sargassum horneri extracts exerting antioxidative activity via the regulation of MAPK/Nrf2/HO-1 machinery,” which we request you to reconsider for publication as an Article in Antioxidants.

As respected reviewer 3 and reviewer 4 suggested, we revised our manuscript point-by-point basis. Please see below and confirm our responses to reviewers’ comments.

The revised manuscript has not been published elsewhere and is not under consideration by another journal. We have approved the manuscript and agree with submission to Antioxidants. There are no conflicts of interest to declare.

We believe that the findings of this study are relevant to the scope of your journal and will be of interest to its readership. The manuscript has been carefully reviewed by an experienced editor whose first language is English and who specializes in editing papers written by scientists whose native language is not English.

We look forward to hearing from you at your earliest convenience.

Sincerely,

Sang H. Lee

Sang-Han Lee, Ph.D.

Professor of Food Enzyme Biotechnology

Graduate School of Food Science and Biotechnology

Kyungpook National University

Daegu 41566

Republic of Korea

Phone No: +82-53-950-7754

Fax No: +82-53-950-6772

Email Address: [email protected]

Reviewer’s comments and responses

Reviewer: 3

1. Perhaps it would make sense to do a statistical comparison (ANOVA) of the experimental and predicted results, and letter-mark the similarity of the means. Because the term 'similar' is vague

Response: Thank you for your insightful comments. We have replaced the text as per reviewer suggestion. Please refer to lines 208-222.

2. As can be seen in Table 2S, not all variables influenced the responses, for example D-frequency was not significant (p>0.05) in all cases etc. Describe results in more details.

Response: Thank you for your valuable comments. We have updated the text and add the necessary details in the text.

3. Add units for parameters in Table 3S.

Response: Thank you for meaningful comment. We have added the units in the Table.

4. I don't quite understand the authors' approach used, it seems to me that there should be final 2 optimized models: the RSM-ANOVA model and the ANN model. Both optimized solutions should be given in Table 4S, and then validated (comparison of predictive and experimental results); then it can be finalized to show which model is better.

Response: Thank you for kind suggestion. We agreed with the reviewer’s comments and in this study, we first compared two model (RSM and ANN) then optimized the condition and then validated. In the old version, ANN alone cannot give the optimized condition (which the reviewer ask), so in the new version, we have added the ANN condition in the Table 5S using the genetic algorithm extension of ANN.

5. This value (43.6) appears to differ significantly from that in Table 4S (23.7), so has the model been validated correctly?

Response: Thank you for important and valuable comment. Yes, because when we performed the RSM and ANN model, the software gave the predicted optimized condition which is when tested, the results were different, that’s why we performed the validation experiments to confirm the both the models are accurate or not.

6. 'first time in S. horneri'

Response: Thank you for important observation. We apologize for the mistake and now we have corrected it also rechecked the whole text to fix this mistake.

7. Shorten the subsections of metabolite identification significantly (after all, this is only a tentative identification), there is no point in rewriting into the text the formulas and MS ions that are given in Table 2. Give metabolites previously reported in the genus Sargassum/ possibly other brown algae, as well as metabolites reported 1st time in S. horneri. This comment refers to my comment on Table 2 from review report 1.

Response: Thank you for important comment. We have tried to remove all the unnecessary text from the heading 3.2.1 as per reviewer’s suggestion.

8. Unclear, re-write.

Phenolic compounds are divided into simple phenolics (including phenolic acids etc) and polyphenols (including flavonoids etc).

Response: Thank you for valuable comment. We have changed the text and modified it accordingly.

9. change to '[M-H]-, m/z'

Response: Thank you for valuable observation. We have changed the text according to reviewer’s suggestion.

10. Include some information about significance of influence of test independent factors on responses (according to ANOVA not all were significant).

Response: Thank you for important and valuable comment. We have added necessary information in the text as per reviewer’s suggestion (lines 462-471).

11. coef. of determination

Response: Thank you for kind suggestion. We apologize for our negligence, we have now replaced the text accordingly.

Reviewer 4 Report

This manuscript discusses the properties of metabolites extracted from the seaweed Sargassum horneri with respect to a specific cell line.

It is divided into three main parts, extraction, testing the antioxidants properties, molecular docking.

Reading the entire manuscript some parts appear more sounding and better investigated than others, the docking part lacks any single data to replicate the simulations done by the authors.

I already ask to the authors to add this information in my previous report:

<Docking is also useless without knowing the detail of the analysis, e.g. ionic strength, water, pH, metals, forcefield and so on.

Response: Thank you for your valuable suggestion. During molecular docking, we usually consider the active pocket site of the protein, binding energy between the ligand and the protein, and the type of interaction (hydrogen bonding, pi-pi interaction, etc.) between the ligand and surrounding amino acids, etc. (please refer to lines 558–562). Before conducting the analysis, we also prepare the ligand and protein by removing the unnecessary moieties, i.e., water molecules. Moreover, it also depends from software to software. Please refer to the following paper: https://doi.org/10.3390/molecules200713384. For calculating the water, pH, and force field, usually SAR analysis and YASARA software might be used, but in our study, we did not use those software devices.>

The answer is completely vague and gives no clue about simulations, suggesting the reviewer refer to a separate review on the subject. However, this approach fails to address the core issue, as none has been added to the supplementary material. Simulations are known to be deeply dependent on the methodology and boundaries applied. Therefore, a reader interested in the topic cannot replicate these simulations. This renders this part meaningless, as we are forced to take the results for granted without the opportunity to judge their validity.

My suggestion to the Editor is to reject this manuscript.

See above

Author Response

May 9, 2024

Prof. Alessandra Napolitano

Editors-in-Chief

Antioxidants

Dear Editor-in-Chief:

Please find enclosed our revised manuscript titled “RSM and ANN-based multifrequency ultrasonic extraction of polyphenol-rich Sargassum horneri extracts exerting antioxidative activity via the regulation of MAPK/Nrf2/HO-1 machinery,” which we request you to reconsider for publication as an Article in Antioxidants.

As respected reviewer 3 and reviewer 4 suggested, we revised our manuscript point-by-point basis. Please see below and confirm our responses to reviewers’ comments.

The revised manuscript has not been published elsewhere and is not under consideration by another journal. We have approved the manuscript and agree with submission to Antioxidants. There are no conflicts of interest to declare.

We believe that the findings of this study are relevant to the scope of your journal and will be of interest to its readership. The manuscript has been carefully reviewed by an experienced editor whose first language is English and who specializes in editing papers written by scientists whose native language is not English.

We look forward to hearing from you at your earliest convenience.

Sincerely,

Sang H. Lee

Sang-Han Lee, Ph.D.

Professor of Food Enzyme Biotechnology

Graduate School of Food Science and Biotechnology

Kyungpook National University

Daegu 41566

Republic of Korea

Phone No: +82-53-950-7754

Fax No: +82-53-950-6772

Email Address: [email protected]

Reviewer’s comments and responses

Reviewer: 4

This manuscript discusses the properties of metabolites extracted from the seaweed Sargassum horneri with respect to a specific cell line.

It is divided into three main parts, extraction, testing the antioxidants properties, molecular docking.

Reading the entire manuscript some parts appear more sounding and better investigated than others, the docking part lacks any single data to replicate the simulations done by the authors.

I already ask to the authors to add this information in my previous report:

<Docking is also useless without knowing the detail of the analysis, e.g. ionic strength, water, pH, metals, forcefield and so on.

Response: Thank you for your valuable suggestion. During molecular docking, we usually consider the active pocket site of the protein, binding energy between the ligand and the protein, and the type of interaction (hydrogen bonding, pi-pi interaction, etc.) between the ligand and surrounding amino acids, etc. (please refer to lines 558–562). Before conducting the analysis, we also prepare the ligand and protein by removing the unnecessary moieties, i.e., water molecules. Moreover, it also depends from software to software. Please refer to the following paper: https://doi.org/10.3390/molecules200713384. For calculating the water, pH, and force field, usually SAR analysis and YASARA software might be used, but in our study, we did not use those software devices.>

The answer is completely vague and gives no clue about simulations, suggesting the reviewer refer to a separate review on the subject. However, this approach fails to address the core issue, as none has been added to the supplementary material. Simulations are known to be deeply dependent on the methodology and boundaries applied. Therefore, a reader interested in the topic cannot replicate these simulations. This renders this part meaningless, as we are forced to take the results for granted without the opportunity to judge their validity.

My suggestion to the Editor is to reject this manuscript.

Response: Thank you for your important and critical suggestion. We have provided the information in the text (lines 179-191). According to the reviewer’s suggestion “Simulations are known to be deeply dependent on the methodology and boundaries applied”, we agreed with the reviewer’s comment, but docking software’s have also some certain boundaries, in the docking software, we tried to recheck that either pH, water, force field can be applicable. But after the reviewing the literature (we even cited the reference), these parameters cannot be checked in the docking. We provided the necessary information and these simulation/experiments can be regenerated by following the methodology (where we have provided the information) specifically lines 186-190 where we have provided the grid box information (please refer to the following papers: https://doi.org/10.1016/j.arabjc.2022.104414; doi: 10.1016/j.fct.2019.110758). Moreover, now we also provide raw docking data which can be used to validate, recheck and replicate the data. If further validation information is required, we are open to discuss and provide the necessary data.

Round 3

Reviewer 3 Report

The 3rd version of manuscript is improved and can be published.

I have found 2 minor comments, see attached pdf document.
